

# Temporal and spatial differences in the vaginal microbiome of Chinese healthy women

Limin Du[1,*], Xue Dong[2,*], Jiarong Song[1], Tingting Lei[3], Xianming Liu[4], Yue Lan[1], Xu Liu[1], Jiao Wang[1], Bisong Yue[1], Miao He[5], Zhenxin Fan[1] and Tao Guo[6]

[1] Sichuan University, Key Laboratory of Bioresources and Ecoenvironment (Ministry of Education), College of Life Sciences, Chengdu, China

[2] Ambulatory Surgery Department, West China Second Hospital, Sichuan University, Chengdu, China

[3] Suining Municipal Hospital of Traditional Chinese Medicine, Suining, Sichuan, China

[4] Mianyang Tumor Hospital, Sichuan Province, Mianyang, China

[5] Institute of Blood Transfusion, Chinese Academy of Medical Sciences, Chengdu, China

[6] Department of Gynecology and Obstetrics, West China Second Hospital, Sichuan University, Chengdu, China

[*] These authors contributed equally to this work.

Corresponding authors
Zhenxin Fan, zxfan@scu.edu.cn
Tao Guo, ivan.gt@163.com

## ABSTRACT

**Background**. Up the reproductive tract, there are large differences in the composition of vaginal microbes. Throughout the menstrual cycle, the structure of the vaginal microbiome shifts. Few studies have examined both in combination. Our study was designed to explore trends in the microbiome of different parts of the vagina in healthy women over the menstrual cycle.

**Methods**. We performed metagenomic sequencing to characterize the microbiome differences between the cervical orifice and mid-vagina throughout the menstrual cycle.

**Results**. Our results showed the vaginal microbiome of healthy women in the cervical orifice and the mid-vagina was similar during the periovulatory and luteal phases, with *Lactobacillus* being the dominant bacteria. In the follicular phase, *Acinetobacter* was detected in the cervical orifice. From the follicular phase to the luteal phase, the community state types (all five community status types were defined as CSTs) in samples No. 10 and No. 11 changed from CST III to CST I. In addition, the composition of the vaginal microbiome in healthy women from different regions of China was significantly different. We also detected viruses including *Human alphaherpesvirus 1* (HSV-1) during periovulatory phase.

**Conclusion**. This study is valuable for understanding whether the microbial composition of the vagina is consistent in different parts of the menstrual cycle.

# INTRODUCTION

The microbiota colonizing the normal vagina is mainly composed of bacteria, fungi, archaea and viruses (*Ma, Forney & Ravel, 2012*). Compared with the upper reproductive

tract (uterus, fallopian tubes and ovaries), the lower reproductive tract (vagina and cervix) of healthy women of reproductive age has more microbes and lower species diversity (*Łaniewski, Ilhan & Herbst-Kralovetz, 2020*; *Ravel et al., 2011*). More than 250 types of bacteria are detected in the vagina, with *Lactobacillus crispatus* and *L. iners* predominating in most healthy women (*Li et al., 2012*). The vaginal microbiome appears to play an important role in the prevention of genitourinary diseases such as urinary tract infections, bacterial vaginosis (BV), human papillomavirus infections, and other sexually transmitted infections (*Borgogna et al., 2020*; *Fredricks, Fiedler & Marrazzo, 2005*; *Zheng et al., 2021*).

The viral component of the human microbiome is called the human virome. In the ano-genital area of healthy women, papillomavirus is the most prevalent eukaryotic virus family (*Gupta, Singh & Goyal, 2020*). Studies of the composition of the vaginal virome in non-pregnant healthy women have the potential to provide a more complete picture of the mechanisms of vaginal microecological health and disorders (*Siqueira et al., 2019*).

A variety of internal and external factors, including race, menstrual cycle, microbial habitat site, hygiene practices, and contraceptive methods, can influence the composition and stability of the vaginal microbiome (*Balle et al., 2020*; *Chen et al., 2017*; *Vodstrcil et al., 2013*). At present, there are few and limited studies on the effect of menstrual cycle on vaginal microbiome. *Song et al. (2020)* used 16S rRNA amplicon sequencing to track the daily vaginal microbiome of young healthy women with different lifestyles and found that during menstruation, vaginal microbial diversity increases, accompanied by a decrease in *Lactobacillus* and an increase in the rate of community change. Their research is mainly at the genus level and lacks the research of other bacteria genera besides *Lactobacillus*, so the results have great limitations. Based on 16S rRNA amplicon sequencing, *Alonzo Martínez et al. (2021)* explored changes in their volunteers' vaginal microbiome over the course of a month and found that 75% of the volunteers maintained the original bacterial community. Amplification sizes for sequencing by these techniques are limited, and species-level identification is considered approximate. Our study considered not only the menstrual cycle, but also different sites (cervical orifice and mid-vagina), and the range of species analyzed was in the top ten, which can be accurately annotated to the species level, greatly compensating for the shortcomings of previous studies.

Due to the limitations of culture-based techniques, the high-throughput sequencing techniques are applied to study the microbiome composition in the vagina, and the most common of which is 16S rRNA amplicon sequencing analysis (*Gajer et al., 2012*; *Hickey et al., 2013*; *Song et al., 2020*). Due to low resolution and limited functional analysis at the species level, studies based on 16S rRNA amplicon sequencing technology have some limitations (*Hillmann et al., 2018*). Metagenomic sequencing has been successfully applied to study the vaginal microbiome to solve the above problems (*Liu et al., 2021*). In recent years, a few studies have turned to metagenomic sequencing to study the species composition and gene function of the vaginal microbiome but these studies have not taken into account the menstrual cycle and vaginal site (*Mancabelli et al., 2021*; *Yang et al., 2020*). However, our study also used metagenomic sequencing and analyzed the vaginal microbiome by site and cycle, which is a complement and improvement to the existing literature.

In this article, we conducted high-throughput metagenomic sequencing of the cervical orifice and the mid-vagina swabs from three time points in the menstrual cycle (follicular, periovulatory, and luteal phases) of healthy women in Chengdu, Sichuan Province, China. We analyzed the composition and functional differences in vaginal microbiome and virome. We explored changes in vaginal microbiome over the menstrual cycle based on sequencing of samples from follicular, periovulatory, and luteal phases. At the same time, we downloaded and analyzed the publicly available datasets from Shenzhen and Hangzhou of China, tring to obtain the geographical differences of vaginal microbiome. These studies that we have done are important for a comprehensive understanding of the structure and changes in the vaginal microbiome.

## METHODS

### Sample collection

In this study, we collected cervical orifice swabs labeled HPVP and mid-vagina swabs labeled MV from volunteers in three periods which include follicular phase labeled A, periovulatory phase labeled B, and luteal phase labeled C. We collected three HPVP samples and 3 MV samples from each volunteer according to follicular phase, periovulatory phase and luteal phase, respectively. To ensure access to sterile swabs, participants were asked to arrive at the hospital each cycle for a vaginal swab sample to be collected by a designated gynecologist, who placed the sample in a labeled sterile tube and kept the sample at $-80$ °C for further analysis (Table S1). All participants provided written informed consent and the study was approved by the Ethics Committee of West China Second Hospital, Sichuan University (2022-054). All internal samples were derived from 11 non-pregnant, reproductive-age women (ages 20 to 30 years) who were recruited from Sichuan University and West China Hospital. Vaginal microecology was detected to ensure that the participating volunteers' vaginas were healthy before the volunteers were enrolled and the following subjects were excluded from the study: (1) women who were pregnant, breastfeeding; (2) women who used antibiotics or vaginal antimicrobials within the previous 30 days; (3) women who had vaginal intercourse within the last week; (4) history of vaginitis, BV, candidiasis, urinary tract infections in the past month.

All metagenomic datasets used in the study included 55 selected publicly available health datasets (*Li et al., 2018*; *Yang et al., 2020*) downloaded from the National Center for Biotechnology Information Sequence Read Achieve (SRA, http://www.ncbi.nlm.nih.gov/sra) and our own internal sequencing datasets. The 55 public datasets included 30 samples from Shenzhen (cervical mucus drawn from the cervical canal) and 25 samples from Hangzhou (a sterile cotton swab is used to sample the cervical orifice). Accession numbers for public datasets were shown in Table S2.

### DNA isolation, library construction, and metagenomic sequencing

The total DNA of vaginal swabs was extracted by Qiagen QIAmp DNA Microbiome Kit (Qiagen, Hilden, Germany). Accurate quantification of DNA concentration was performed using Qubit® DNA Assay Kit (Invitrogen, Waltham, MA, USA). Subsequently, a complete library was produced through the steps of end repair, adding A-Tail, adding sequencing

connectors, purification, and PCR amplification. The DNA library was further sequenced on Illumina platform, and we obtained clean data using KneadData (v0.6.1) to remove host contamination in raw data and using Trimmomatic (v0.36) (*Bolger, Lohse & Usadel, 2014*) to remove adapters and low-quality reads in raw data. MEGAHIT (v1.2.9) (*Li et al., 2015*) was used to *de novo* assemble the target reads to obtain consecutive long fragments contigs (–min-contig-len 300). Contigs were used for subsequent taxonomic identification and gene function analysis. For the 55 public data downloaded, we carried out the same operation.

## Taxonomic classification of sequence reads

Kraken2 (v2.1.3) (*Ye et al., 2019*) was used to classify the contigs quickly and accurately by mapping to an officially built standard database. Linear discriminant analysis effect size (LEfSe) (*Segata et al., 2011*) was used to identify species differences in taxa and look for species markers in R version 4.3.1. Principal coordinate analysis (PCoA) was performed to evaluate similarities or dissimilarities in the composition of the study sample communities based on Bray–Curtis distances in R version 4.3.1.

## Gene function analysis of metagenomes

The metagenomic genes of all samples were predicted based on Prodigal (v2.6.3) (*Hyatt et al., 2010*). CD-HIT (v4.8.1) was used to build non-redundant gene sets (*Fu et al., 2012*). The non-redundant gene sets were translated into amino acid sequences. All amino acid sequences were aligned in the Carbohydrate-Active enZYmes (CAZy) database using DIAMOND (v2.1.8) (*Buchfink, Xie & Huson, 2015*; *Lombard et al., 2014*). Gene family abundances and microbiome metabolic pathways were evaluated using HUMANn3 (v3.7) based on ChocoPhlAn database and UniRef90 EC database (*Franzosa et al., 2018*). To identify the abundance of antibiotic resistance genes (ARGs) in the vaginal microbiome, ARGs were quantified using ShortBRED (v0.9.3) based on the comprehensive antibiotic resistance database (CARD) (*Kaminski et al., 2015*).

## Virome analysis

Viral sequencing was conducted in Chengdu Life Baseline Technology Co., Ltd. Samples were centrifuged at 2,500× g for 5 min, then the supersolution was centrifuged again and passed through a filter to remove debris and cells. The samples were treated with 2 ul lysozyme at room temperature for 30 min, then continued with 0.2 times the volume chloroform under RT for 10 min. Then 10U Tubro DNase I and 2 ug RNase A were added to the new centrifuge supersolution, subsequently thermally deactivated at 65 °C for 10 min. The Qiagen MinElute virus and Qubit dsDNA HS Assay kit were used to extract and quantify the virus DNA. Small fragment libraries of 200–500 bp were constructed by standard or microlibrary construction, and paired 150 bp reads were sequenced.

## RESULTS

We recruited 14 volunteers in total, and the samples from three of them were excluded due to one of them was diagnosed with HPV infection, and the remaining two samples

were contaminated (5, 6 and 12). Only 59 samples were collected in the end due to the incomplete cervical orifice samples of individuals (1, 7 and 9 among the 11 volunteers. We then grouped these samples by sampling site and sampling period. In addition, our initial analyses found that the pathogenic bacterium *Gardnerella* was dominant in the vaginal microbiome of No. 9 and No. 14 individuals, thus we subsequently removed the two samples from downstream analyses and studied them separately (File S1).

## Vaginal microbial composition in different sites of the same period

The results of species classification and identification showed that the main members of the vaginal microbiome were bacteria. Firmicutes (90.15–99.98%) was identified as the most abundant phylum in both cervical orifice and mid-vagina (Fig. S1A). The relative abundance of Firmicutes (96.81–99.93% in the cervical orifice, 98.79–99.98% in the mid-vagina) (Wilcoxon's rank-sum test, $p < 0.01$) was lower, while the relative abundance of Proteobacteria (0.03–3.15% in the cervical orifice, 0.01–0.21% in the mid-vagina) (Wilcoxon's rank-sum test, $p < 0.05$) was higher in the cervical orifice, during the period A (Fig. S1B). The genus *Lactobacillus* (90.16–99.97%), was dominant in the vaginal microbiome, in both cervical orifice and mid-vagina (Fig. S1C). At the species level, *L. crispatus* and *L. iners* were the top two abundant species in both two sites (Figs. 1A, 1B, 1C). In addition, low relative abundance of the opportunistic pathogen species *Acinetobacter baumannii* (0–0.09%) (Wilcoxon's rank-sum test, $p < 0.05$) was identified in the cervical orifice during period A (Fig. S1D).

A statistically significant PCoA plot based on a phylum-level relative abundance during period A showed that there were significant differences in vaginal microbiome between the cervical orifice and the mid-vagina ($p < 0.05$) (Fig. 1D). These differences were not present in other groups or at other taxonomic levels. We found that Firmicutes was enriched in the mid-vagina whereas Proteobacteria and *Acinetobacter* were enriched in the cervical orifice during period A based on LEfSe analysis (Figs. 1E and S1E). Interestingly, no taxonomic differences in vaginal microbiome were found between the cervical orifice and the mid-vagina during periods B and C.

## Vaginal microbial composition in different cycles of the same site

Firmicutes was the dominant phylum during periods A, B and C (Fig. S2A). Consistent with the phylum level analysis, the relative abundance of *Lactobacillus* in Firmicutes was the highest during periods A, B and C (Fig. S2B). Similarly, at the species level, *L. crispatus* and *L. iners* were the top two species in three periods (Fig. 2A).

The results of PCoA analysis at the phylum level and genus level in the cervical orifice showed that there were significant differences in the samples from different periods ($p < 0.05$) (Figs. 2B, 2C). In addition, all samples were not separated by the period at the three taxonomic levels in the mid-vagina. Further, LEfSe results demonstrated that differential markers were identified, with Firmicutes enriched during period C and Proteobacteria enriched during period A in the cervical orifice (LDA > 3) (Fig. S2C). Such differences also appeared at the genus level. For instance, *Lactobacillus* was enriched during period C while *Pseudomonas* and *Acinetobacter* were enriched during period A in

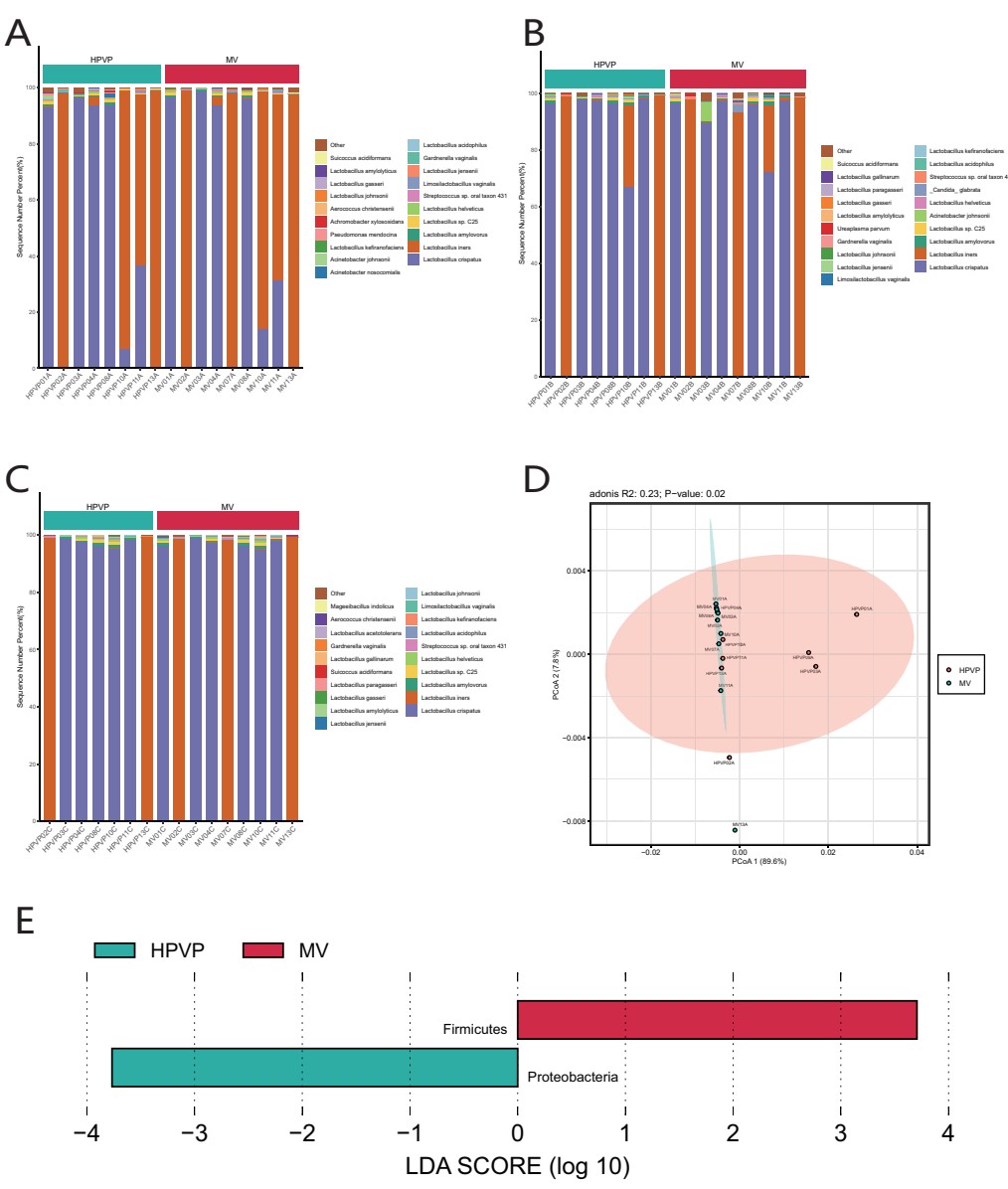

**Figure 1 The microbiome composition of the cervical orifice and the mid-vagina.** (A) Top 20 most abundant bacterial species in two different sites of vagina during period A. (B) Top 20 most abundant bacterial species in two different sites of vagina during period B. (C) Top 20 most abundant bacterial species in two different sites of vagina during period C. (D) PCoA plot based on Bray–Curtis distance of phylum-level relative abundance profile during period A. (E) The difference of phylum-level relative abundance between the cervical orifice and the mid-vagina during period A by LEfSe ($p < 0.05$ and LDA > 3).

the cervical orifice (LDA > 3) (Fig. 2D). To explore the changes of relative abundances during periods A, B, and C, some important taxa from above results, including two phyla (Firmicutes and Proteobacteria), two genera (*Lactobacillus* and *Acinetobacter*), and two species (*L. crispatus* and *L. iners*) were selected for further analysis. The relative

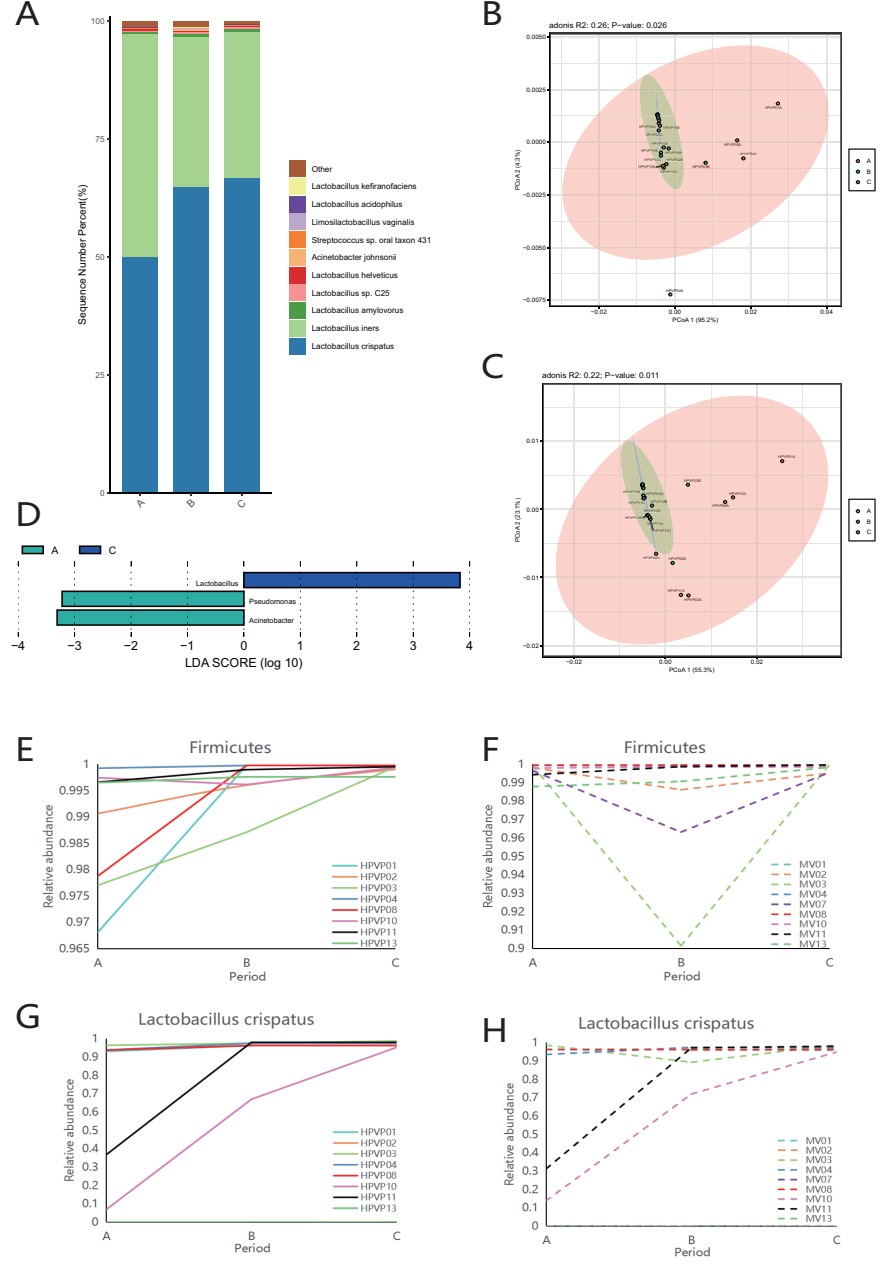

**Figure 2 Vaginal microbiome composition during periods A, B and C.** (A) Top 10 most abundant bacterial species of vagina in three different periods. (B) PCoA plot based on Bray–Curtis distance of phylum-level relative abundance profile in the cervical orifice. (C) PCoA plot based on Bray–Curtis distance of genus-level relative abundance profile in the cervical orifice. (D) The difference of genus-level relative abundance between periods A, B and C in the cervical orifice by LEfSe ($p < 0.05$ and LDA > 3). (E) Dynamics of the relative abundance of Firmicutes in the cervical orifice throughout the menstrual cycle. (F) Dynamics of the relative abundance of Firmicutes in the mid-vagina throughout the menstrual cycle. (G) Dynamics of the relative abundance of *L. crispatus* in the cervical orifice throughout the menstrual cycle. (H) Dynamics of the relative abundance of *L. crispatus* in the mid-vagina throughout the menstrual cycle.

**Table 1  Community state types (CSTs) of vaginal microbiome in different sites of vagina or different periods.**

|         | Cervical orifice (%) | Mid-vagina (%) | Period A (%) | Period B (%) | Period C (%) |
|---------|---------------------|----------------|--------------|--------------|--------------|
| CST I   | 15 (65.22)          | 16 (59.26)     | 8 (47.06)    | 12 (70.59)   | 11 (68.75)   |
| CST II  | 0                   | 0              | 0            | 0            | 0            |
| CST III | 8 (34.78)           | 11 (40.74)     | 9 (52.94)    | 5 (29.41)    | 5 (31.25)    |
| CST IV  | 0                   | 0              | 0            | 0            | 0            |
| CST V   | 0                   | 0              | 0            | 0            | 0            |

abundance of Firmicutes in almost all samples gradually increased throughout the cycle, whereas Proteobacteria showed the opposite trend in the cervical orifice (Figs. 2E and S2D). Consistently, the relative abundance of *Lactobacillus* generally increased throughout the cycle (Fig. S2E). In addition, the relative abundance of *Acinetobacter* in the cervical orifice showed a downward trend throughout the cycle (Fig. S2F). Different from the cervical orifice, the relative abundance of Firmicutes and *Lactobacillus* first decreased and then increased in the mid-vagina from one-third of the samples (Figs. 2F and S2G). An unexpected result showed that the relative abundance of *L. crispatus* increased significantly while the relative abundance of *L. iners* decreased significantly throughout the cycle in samples No. 10 and No. 11 (Figs. 2G, 2H and S2H, S2I).

## Community state type transitions

Table 1 showed the classification of five previously reported community state types (CSTs). *Ravel et al. (2011)* divided the vaginal microbiota into five community state types (CSTs) based on the relative abundance of dominant species of Lactobacillus: CST I dominated by *L. crispatus*, CST II dominated by *L. gasseri*, CST III dominated by *L. iners*, CST IV with a lower proportion of Lactobacillus and a higher proportion of anaerobic bacteria, and CST V dominated by *L. jensenii*. We found that 15/23 (65.2%) in the cervical orifice and 16/27 (59.3%) in the mid-vagina were type CST I, 8/23 (34.8%) in the cervical orifice and 11/27 (40.7%) in the mid-vagina were type CST III. Samples during periods B and C were mostly assigned to CST I, whereas the samples during period A were mostly assigned to CST III. The results of the cluster heat map showed that the CST of almost all samples remained in a stable state during the whole cycle except for a few individuals with dynamic changes, which the CST in samples No. 10 and No. 11 changed from CST III to CST I (Figs. 3A, 3B).

## Comparison of vaginal microbiome in healthy women from different regions

We extracted the cervical orifice samples of period B in our study, which was the closest period to the downloaded public sequence. Subsequently, we compared the samples from Chengdu, Hangzhou, and Shenzhen to study the regional differences of the vaginal microbiome.

We observed that there were great differences in vaginal microbiome of healthy women between the three regions at the levels of phylum, genus, and species (Figs. 4A, 4B, 4C). The results of PCoA exhibited that the samples from Chengdu, Hangzhou, and

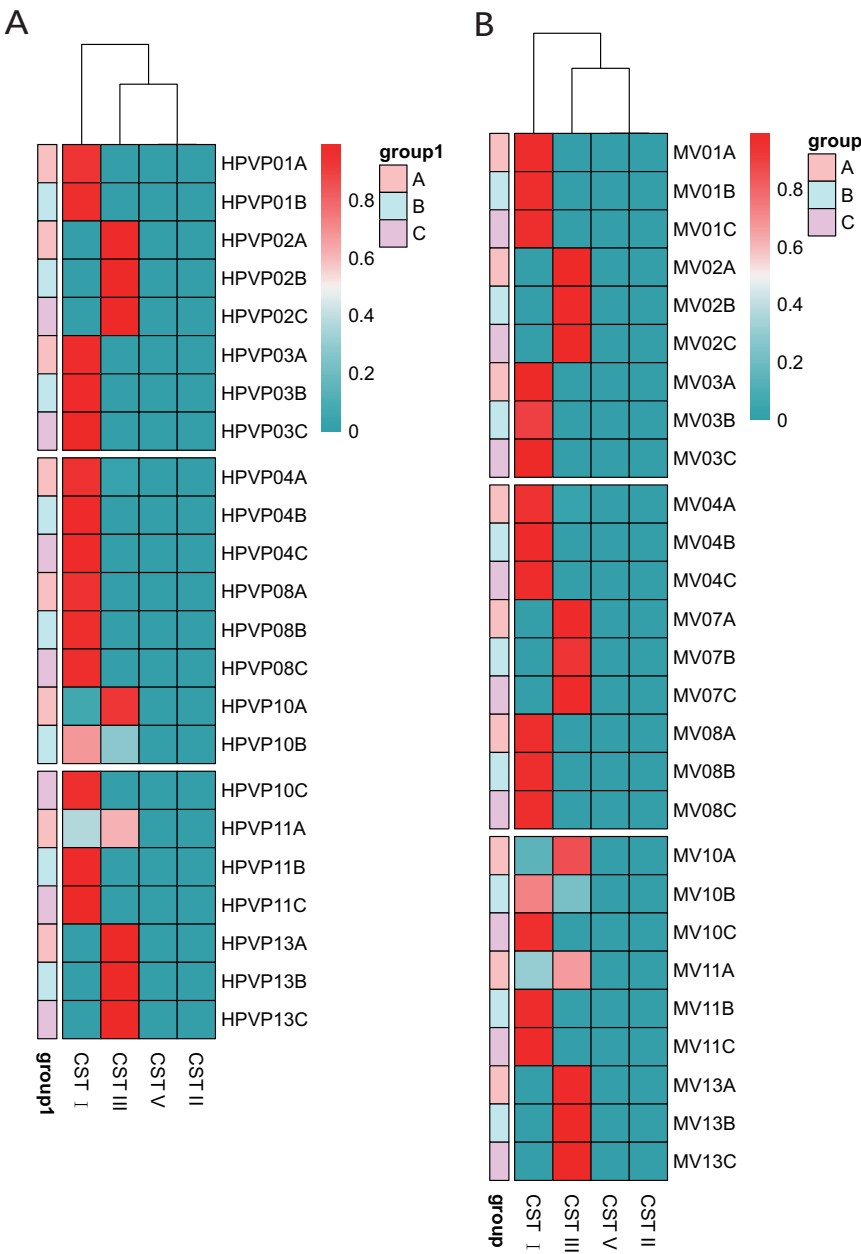

**Figure 3  The distribution of the CSTs during three periods. The redder the color, the higher the relative abundance.** (A) The distribution of the CSTs during three periods in the cervical orifice. (B) The distribution of the CSTs during three periods in the mid-vagina.

Shenzhen could be significantly separated at the phylum, genus and species levels (R > 0, $p < 0.01$) (Figs. 4D, 4E, 4F). The LEfSe results demonstrated that only Firmicutes was significantly enriched in the vaginal microbiome of women in Chengdu (LDA > 3) (Fig. 4G). The most differentially abundant genera in the vaginal microbiome of women in Shenzhen, Hangzhou, and Chengdu were *Gardnerella*, *Staphylococcus* and *Lactobacillus*,

respectively (Fig. 4H). Meanwhile, we found that five pathogenic microorganisms including *Gardnerella vaginalis, A. johnsonii, Chlamydia trachomatis, Mycoplasma genitalium* and *Sneathia vaginalis* were enriched in the vaginal microbiome of women in Shenzhen, and *Staphylococcus_epidermidis*, another pathogenic microorganisms was enriched in the vaginal microbiome of women in Hangzhou (LDA > 3) (Fig. 4I).

## Functional characteristics of the vaginal microbiome

The quantitative results of microbial gene families and metabolic pathways showed that the microbiome in both the cervical orifice and the mid-vagina was mainly enriched in the glycolysis IV pathway and the 6-hydroxymethyl-dihydropterin diphosphate biosynthesis III pathway (LDA > 3) (Fig. 5A). In addition, GH13 of glycoside hydrolase (GH) family was enriched in two sites of vagina during period A, and the expression of GT26 belonging to the glycosyl-transferase (GT) family was higher in the mid-vagina than in the cervical orifice during period C (LDA > 3, $p < 0.05$) (Figs. 5B, 5C, 5D). Quantitative results of ARGs to vaginal microbiome revealed that *tet* M gene was only enriched in period A in two sites (LDA > 3) (Fig. 5E).

To compare the functional and metabolic differences of the vaginal microbiome in different regions in China, we performed the same metagenomic sequencing operation on downloaded public sequences. The carbohydrate enzyme activity was significantly lower in the vaginal microbiome of women in Shenzhen than that of the other two regions (LDA > 3) (Fig. 5F). Furthermore, the macrolide antibiotic tetracycline resistance *Erm*X gene was significantly enriched in the vaginal microbiome of women in Hangzhou (LDA > 3) (Fig. 5G). In contrast, women from the other two regions had lower expression of ARGs. Three metabolic pathways, including coenzyme A biosynthesis II, UMP biosynthesis and adenosine ribonucleotides de novo biosynthesis, were enriched in the vaginal microbiome of women in Chengdu. In contrast, the vaginal microbiome of women in the other two regions was primarily enriched in pyruvate fermentation to acetate and (S)-lactate I ($p < 0.01$) (Fig. 5H).

## The virome in the vaginal microbiome

Uroviricota was the phylum with the highest relative abundance in the cervical orifice during periods A, B, and C (Fig. 6A). *Slopekvirus* and *Whackvirus* were the top two genera in the relative abundance of the cervical orifice during periods A and C, while *Kagunavirus* and *Pahexavirus* were the two genera during period B (Fig. 6B). The LEfSe result showed that 36 genera were enriched in the cervical orifice during period B, while only one or two genera were enriched during periods A and C (LDA > 3) (Fig. 6C). At the species level, *Klebsiella virus Matisse* and *Klebsiella virus KP27* were enriched in periods A and C, while *uncultured crAssphage* and *Antheraea pernyi nucleopolyhedroviru* were enriched during period B (LDA > 3) (Figs. 6D, 6E).

We performed gene function annotation analysis for vaginal virome and found that seven metabolism-related pathways were enriched during period B, while only adenosine ribonucleotides de novo biosynthesis pathway and guanosine ribonucleotides *de novo* biosynthesis pathway were enriched during period A, and no significantly enriched

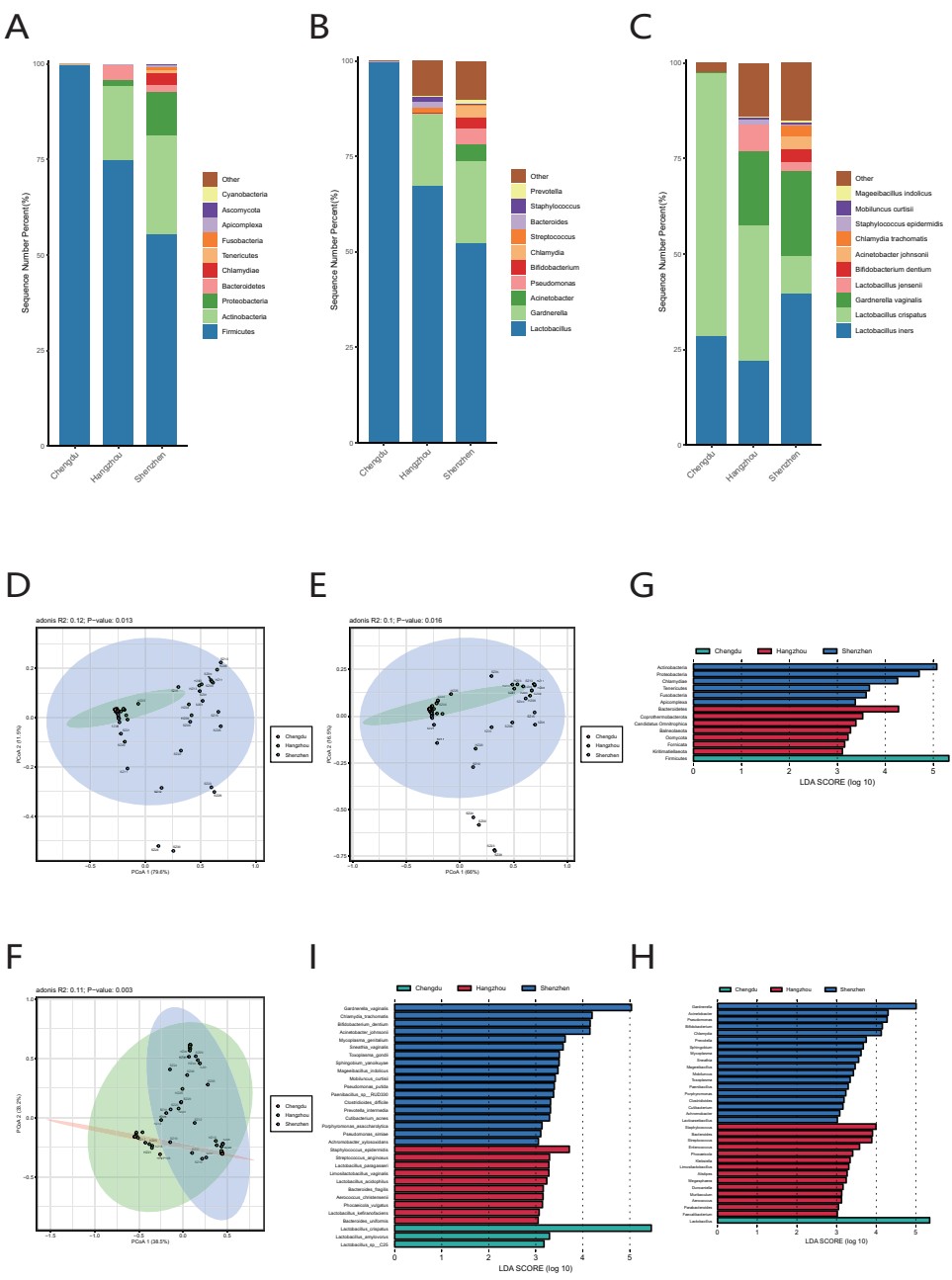

**Figure 4** Comparison of vaginal microbiome of healthy women in Chengdu, Shenzhen and Hangzhou.
(A) Top 10 most abundant bacterial phyla in the cervical orifice of healthy women in three regions. (B) Top 10 most abundant bacterial genera in the cervical orifice of healthy women in three regions. (C) Top 10 most abundant bacterial species in the cervical orifice of healthy women in three regions. (D) PCoA plot based on Bray–Curtis distance of phylum-level relative abundance profile in three regions. (E) PCoA plot based on Bray–Curtis distance of genus-level relative abundance profile in three regions. (F) PCoA plot based on Bray–Curtis distance of species-level relative abundance profile in three regions. (G) The difference of phylum-level relative abundance in three regions by LEfSe ($p < 0.05$ and LDA > 3). (H) The difference of genus-level relative abundance in three regions by LEfSe ($p < 0.05$ and LDA > 3). (I) The difference of species-level relative abundance in three regions by LEfSe ($p < 0.05$ and LDA > 3).

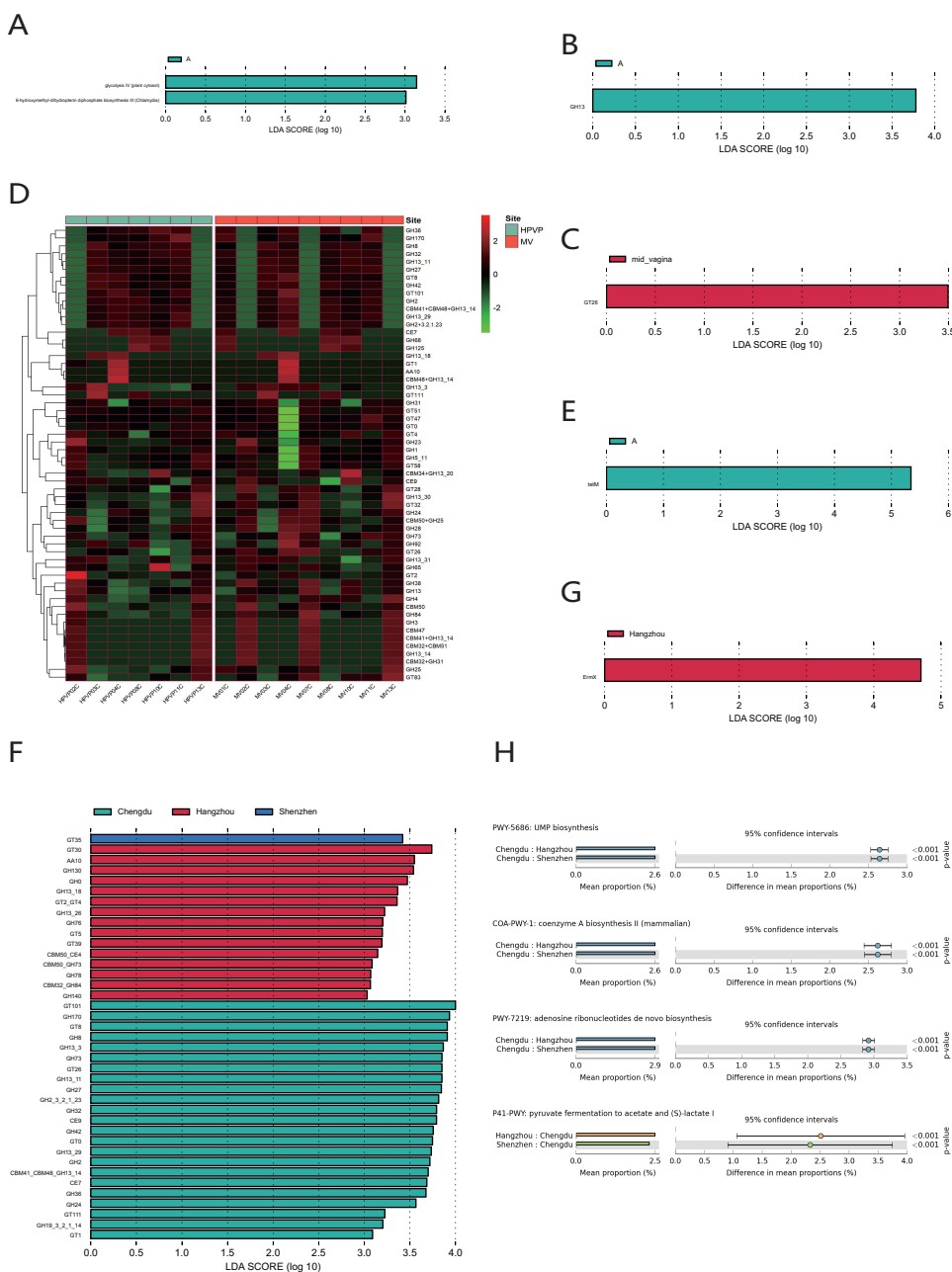

**Figure 5** **Functional analysis of the vaginal microbiome.** (A) Differential analysis of metabolic pathways between periods A, B and C ($p < 0.05$ and LDA > 3). (B) Differential analysis of vaginal microbial CAZymes between periods A, B and C ($p < 0.05$ and LDA > 3). (C) Differential analysis of vaginal microbial CAZymes between the cervical orifice and the mid-vagina during period C ($p < 0.05$ and LDA > 3). (D) The distribution of the vaginal microbial CAZymes between the cervical orifice and the mid-vagina. (E) Differential analysis of vaginal microbial ARGs between periods A, B and C ($p < 0.05$ and LDA > 3). (F) Differential analysis of vaginal microbial CAZymes in the cervical orifice of healthy women in three regions ($p < 0.05$ and LDA > 3). (G) Differential analysis of vaginal microbial ARGs in the cervical orifice of healthy women in three regions ($p < 0.05$ and LDA > 3). (H) Differential analysis of metabolic pathways in the cervical orifice of healthy women in three regions ($p < 0.05$).

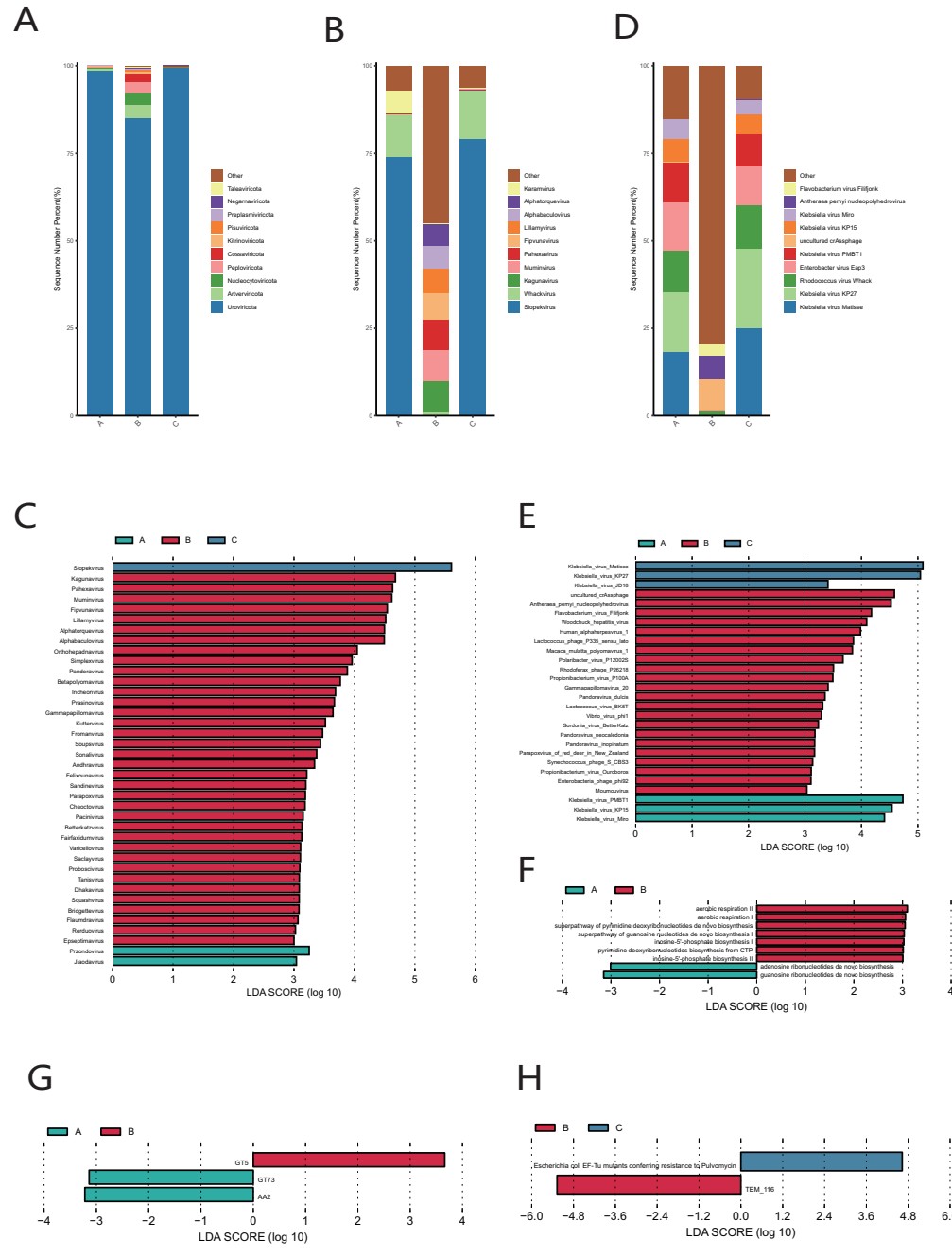

**Figure 6** **Vaginal virome composition during periods A, B and C.** (A) Top 10 most abundant virus phyla in the cervical orifice of healthy women during periods A, B and C. (B) Top 10 most abundant virus genera in the cervical orifice of healthy women during periods A, B and C. (C) Differential analysis of vaginal virus genera ($p < 0.05$ and LDA > 3). (D) Top 10 most abundant virus species in the cervical orifice of healthy women during periods A, B and C. (E) Differential analysis of vaginal virus species ($p < 0.05$ and LDA > 3). (F) Differential analysis of metabolic pathways in the vaginal virome ($p < 0.05$). (G) Differential analysis of CAZymes in the vaginal virome ($p < 0.05$ and LDA > 3). (H) Differential analysis of ARGs in the vaginal virome ($p < 0.05$ and LDA > 3).

metabolic pathway was detected during period C (LDA > 3) (Fig. 6F). Identification and quantification of CAZymes showed that GT5 was significantly more abundant during period B, while GT73 and AA2 were enriched during period A (LDA > 3) (Fig. 6G). The *Escherichia coli* EF-Tu mutants encoding pulvomycin antibiotic resistance was most prevalent in the cervical orifice during period C and the TEM-116 ecoding cephalosporin antibiotic resistance was most prevalent in the cervical orifice during period B (LDA > 3) (Fig. 6H).

## DISCUSSION

Vaginal microbiome varies from individual to individual. The menstruation, diet, and exercise all regulate the composition and stability of the vaginal microbiome and may affect vaginal and reproductive health (*Song et al., 2020*). We studied the differences in vaginal microbiome between the cervical orifice and the mid-vagina in samples with the same sampling period. We found that the composition and structure of microorganisms in the cervical orifice and the mid-vagina during periods B and C were obviously simular, with *Lactobacillus* as the main microflora, which was consistent with the results of previous studies (*Alonzo Martínez et al., 2021*). *Lactobacillus* spp. is important beneficial bacteria in the human vagina (*Tachedjian et al., 2017*). Previous studies have shown that *Lactobacillus* spp. can inhibit the infection and proliferation of pathogenic microorganisms by adhering to the surface of vaginal epithelial cells to build a physical barrier, produce bacteriocin, organic acid, hydrogen peroxide and other antibiotic factors, compete for nutrients, and produce various metabolites such as lactic acid, and thus play an anti-infection and anti-tumor role (*Voravuthikunchai, Bilasoi & Supamala, 2006*). In addition, at the species level, our results indicated that CSTs in both cervical orifice and mid-vagina were CST I, which was considered to be the most stable and clinically important CST with the greatest potential for improving health (*Mancabelli et al., 2021*). In contrast, we observed significant differences between the microbiome of the cervical orifice and mid-vagina during the period A, even though the dominant microflora in both sites was *Lactobacillus*. These differences were specifically manifested in the enrichment of *Acinetobacter* in the cervical orifice during the period A, while Firmicutes of probiotics was concentrated in the mid-vagina. *Acinetobacter* spp. are opportunistic pathogen, which can be transformed into pathogenic bacteria under certain conditions and is usually closely associated with ovarian cancer and vaginitis (*Cazzaniga et al., 2022*; *Nguyen et al., 2022*). The lack of *lactobacillus* in the vagina may be one of the potential causes of *Acinetobacter* infection. *A. baumannii* is a recognized pathogen of nosocomial transmission and infection (*Hamidian & Nigro, 2019*). The whole process of sampling in our study was conducted in the hospital. Although the operation was carried out to avoid infection as much as possible, the abundance of *A. baumannii* in our samples was too low, and there was also the possibility of nosocomial infection. Similarly, previous studies have found the presence of *Acinetobacter* in the stool of babies delivered vaginally (1 or 2 days), which was ultimately interpreted as a nosocomial infection (*Pandey et al., 2012*).

We studied the characteristics of the vaginal microbiome during periods A, B, and C in samples from the same sampling site. Our results showed that *Lactobacillus* had the

highest relative abundance in the mid-vagina during periods A, B and C. In contrast, there were significant differences in the microflora of the cervical orifice during periods A and C. These differences were reflected in the enrichment of *Pseudomonas* and *Acinetobacter* during period A, while the enrichment of *Lactobacillus* during period C. The results of functional annotation revealed that the cervical orifice and the mid-vagina microbiome of period A were rich in the metabolic pathway 6-hydroxymethyl-dihydropterin diphosphate biosynthesis III, which was mainly contributed by *Chlamydia* spp. (*Adams et al., 2014*). Based on our results, we considered that there were individuals in the study who were infected with *Chlamydia*. In addition, *tet* M gene of tetracycline resistance was enriched in period A, which may indicate that the existence of tetracycline antibiotic resistance gene provides convenient conditions for *Chlamydia* spp. infection.

After further study, we also found that the relative abundance of Proteobacteria gradually decreased, while the relative abundance of Firmicutes gradually increased from period A to period C in the cervical orifice. This trend also existed at the genus level, where the relative abundance of *Acinetobacter* gradually decreased, while the relative abundance of *Lactobacillus* gradually increased. The findings revealed once again that probiotics were antagonistic to pathogenic bacteria. In addition, we found that the CSTs of samples No. 10 and No. 11 changed with the menstrual cycle, and the trend of change was from CST III to CST I. Throughout the menstrual cycle vaginal CST change are not uncommon. Previous studies have also pointed out that CST changes over time. For example, Gajer et al.'s study on the vaginal community of healthy women based on 16S sequencing found that during the menstrual cycle, vaginal CST in healthy women changed from type CST I to type CST III, and then changed back to type CST I at the end of menstruation, opposite the direction of CST shift in our results (*Gajer et al., 2012*). In their study this shift was an individual phenomenon, so we attributed this opposite shift phenomenon to individual differences. Studies have revealed significant differences in microbiomes between the secretory and proliferative phases (luteal and follicular phases) of the menstrual cycle regardless of disease condition or body part (*Pelzer et al., 2018*; *Sola-Leyva et al., 2021*). Some studies have pointed out that there is a significant pH gradient in the reproductive tract, and the change of vaginal CST during the menstrual cycle may be closely related to vaginal PH (*Lykke et al., 2021*). In addition, we thought this shift seems to be related to the spread of microorganisms in the mid-vagina as the cervical orifice opens during periovulatory.

The results of our exploration of the vaginal microbiome in Chengdu, Hangzhou, and Shenzhen showed that there were significant differences in the composition of the cervical orifice microbiome in different regions of China. The relative abundance of *Lactobacillus* in cervical orifice microbiome of women in Chengdu was significantly higher. Although pathogenic bacteria were present in the samples from Hangzhou and Shenzhen, *Lactobacillus* was the dominant bacterium. At the species level, the samples from Chengdu were enriched with *L. crispatus*, while the samples from Shenzhen were enriched with *G. vaginalis, A. johnsonii, C. trachomatis* and *M. genitalium*. The samples from Hangzhou were enriched with pathogenic microorganisms *S._epidermidis*. *C. trachomatis* is mainly pathogenic to the urogenital tract (*Cevenini, Donati & Sambri, 2002*). *M. genitalium* is associated with sexually transmitted diseases (*Yueyue et al., 2022*). In addition, members

of the genus *Gardnerella* are usually one of the most abundant bacteria in BV (*Agarwal & Lewis, 2021*). S._epidermidis is now considered an important opportunistic pathogen, the most common source of infection on indwelling medical devices, and it is now the most common cause of hospital-acquired infections (*Otto, 2009*). The differences in the female microbiome between different regions were mainly due to the species and abundance of pathogenic microorganisms, but the abundance of these pathogenic or opportunistic pathogens was much lower than the abundance of *Lactobacillus* and only a few individuals had a high abundance of pathogenic microorganisms. Further analysis results at the functional level showed that the vaginal microbiome of women in Hangzhou was enriched with multiple ARGs, indicating that women in Hangzhou had resistance to vaginal antibiotics. In contrast, women in Chengdu had a healthier vaginal microbiome.

Previous studies of virome analysis of vaginal samples from healthy women have shown that most of the identified viral sequences were from double-stranded DNA phages, with eukaryotic viruses accounting for only 4% of total readings (*Jakobsen et al., 2020*). In addition to anatomical location, there are many factors related to the structure of vaginal virome, such as diet, age, and geographical location of the sampled individual (*Liang & Bushman, 2021*). The dominant virus in our results was bacteriophage. Uroviricota was the most abundant virus during periods A, B and C. The phylum Uroviricota includes crAss-like phages, which are common in human enteroviruses. In addition, we found significant differences in the vaginal virome of periods A, B, and C. In period B, multiple viruses were enriched, and two viruses, uncultured crAssphage, and HSV-1, were detected. Studies have shown that crAssphage is the most abundant virus known to exist in humans and one of the most common phages in publicly available metagenomes (*Dutilh et al., 2014*). HSV-1 virion is the herpes virus, which can establish a latent infection but when reactivated can cause herpes of the skin or genitalia, conjunctivitis, keratitis, encephalitis or herpes eczema (*Kukhanova, Korovina & Kochetkov, 2014*). HSV-1 may also be involved in the pathogenesis of multiple sclerosis (*Bello-Morales et al., 2012*). According to our results, it appeared that healthy women during period B were more susceptible to virus invasion and infection of the cervical orifice.

## CONCLUSIONS

In conclusion, we conducted metagenomic and virome sequencing of the vaginal microbiome in healthy women. Our results showed that the dominant microflora in the vagina of healthy women was *Lactobacillus*, and there was no significant difference in the microbiome composition between the cervical orifice and the mid-vagina during the corresponding cycle. We also found that healthy women living in different regions had significantly different vaginal microbiomes, the reasons and mechanisms leading to this difference still need more and more in-depth research to clarify. Our results also showed the relative abundance of Acinetobacter gradually decreased, while the relative abundance of *Lactobacillus* gradually increased throughout the menstrual cycle. The interaction between probiotics and pathogenic bacteria provides a reference for the preparation of probiotic preparations, and also valuable for the use of clinical antibiotics. Overall, there are also

many limitations to this study. Changes in CSTs occur in a small number of individuals, and small sample sizes make it difficult to ensure that the results obtained are universal. The age range of volunteers in this study was 20–30 years old, and it was not possible to determine the vaginal microbiome of healthy women in other age ranges. Future metagenomic studies with larger sample sizes are needed to reveal more information.

## ACKNOWLEDGEMENTS

All sample collection and vaginal microecological tests were carried out at West China Hospital. We would like to thank the technical support provided by West China Hospital and every volunteer who participated in the study.

### Funding

This research was supported by the Sichuan Science and Technology Program (2023NSFSC1935). The funders had no role in study design, data collection and analysis, decision to publish, or preparation of the manuscript.

### Grant Disclosures

The following grant information was disclosed by the authors:
The Sichuan Science and Technology Program (2023NSFSC1935).

### Competing Interests

The authors declare there are no competing interests.

### Author Contributions

- Limin Du performed the experiments, analyzed the data, prepared figures and/or tables, authored or reviewed drafts of the article, and approved the final draft.
- Xue Dong conceived and designed the experiments, prepared figures and/or tables, authored or reviewed drafts of the article, and approved the final draft.
- Jiarong Song performed the experiments, prepared figures and/or tables, and approved the final draft.
- Tingting Lei performed the experiments, prepared figures and/or tables, and approved the final draft.
- Xianming Liu performed the experiments, prepared figures and/or tables, and approved the final draft.
- Yue Lan performed the experiments, analyzed the data, authored or reviewed drafts of the article, and approved the final draft.
- Xu Liu conceived and designed the experiments, authored or reviewed drafts of the article, and approved the final draft.
- Jiao Wang conceived and designed the experiments, authored or reviewed drafts of the article, and approved the final draft.
- Bisong Yue conceived and designed the experiments, authored or reviewed drafts of the article, and approved the final draft.

- Miao He conceived and designed the experiments, authored or reviewed drafts of the article, and approved the final draft.
- Zhenxin Fan conceived and designed the experiments, authored or reviewed drafts of the article, and approved the final draft.
- Tao Guo conceived and designed the experiments, authored or reviewed drafts of the article, and approved the final draft.

## Human Ethics

The following information was supplied relating to ethical approvals (i.e., approving body and any reference numbers):

This study was approved by the Medical Ethics Committee of West China Second Hospital, Sichuan University(2022-054).

## Data Availability

The data is available at the CNGB: CNP0004917.

## Supplemental Information

Supplemental information for this article can be found online at http://dx.doi.org/10.7717/peerj.16438#supplemental-information.

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
