# Peer review of "Temporal and spatial differences in the vaginal microbiome of Chinese healthy women"

_PeerJ, doi:10.7717/peerj.16438_

## Round 0.1 · original submission · Major Revisions

The manuscript has been assessed by three independent reviewers and I strongly suggest addressing the concerns raised by all three reviewers before your paper could be considered for publication.

1. Introduction needs to be reframed for better understanding and flow.
2. The significance and novelty of the study is highly questionable. The authors are suggested to provide the rationale for the current study.
3. Experimental design needs extensive revision and authors are suggested to provide more details on the sample collection and tools used.
4. Relatively small size is a major concern since it is not enough to conclude the results obtained.
Overall, the manuscript in its present form is not suitable for publication and needs major revision.

·

Basic reporting

The authors collected the vaginal microbiome samples from healthy, non-pregnant, reproductive-age (20—30 years old) women from China and analyzed the vaginal microbiome from two sites (cervical orifice and mid-vagina) across three stages of the menstruation cycle. The collected metagenomic samples were shotgun metagenome sequenced and were assessed for their taxonomic compositions using Kraken2, alongside other analyses using gene identification. The authors claim that the changes in vaginal microbiome in the context of menstruation have not been well-characterized and that their work will attempt at filling in this gap in the literature.
A significant concern with the findings from this work is that the authors claim the detection of Acinetobacter baumannii, which is the highest priority ESKAPE antimicrobial-resistant pathogen listed by WHO for novel drug development. However, this detection appears to be solely dependent on taxonomic identification and classification via Kraken2. It is well-known that the k-mer-based taxonomic classification from shotgun metagenomic sequences (reads) using Kraken2 is convenient for rapid taxonomic classification but may not be highly reliable in all setups (datasets). To make inferences that A. baumannii has been detected, the authors need to perform extensive analysis, including clinical isolation, to ensure the authenticity of this pathogen detection.

Experimental design

It is unclear whether the taxonomic identifications and gene function analyses were performed on the metagenomic reads or assembled prior to subsequent analyses. The read-based and assembly-based approaches have their own share of pros and cons, but this needs to be made more explicit. For this study, it is recommended that the authors perform a metagenomic assembly for each sample if they haven’t done this already, and contrast their findings.
Additionally, the authors need to provide the specific version of each tool used (in Methods) to support the reproducibility and repeatability of their work.

Validity of the findings

1. It is concerning that an ESKAPE pathogen, particularly A. baumannii, was detected in this dataset. The results in the current study rely primarily on Kraken2 results. Inferences made using Kraken2 are known to be erroneous in some cases (particularly in shotgun metagenomic samples where the microbiome may not be well represented in the reference databases) owing to the underlying k-mer matching and modeling. It is, therefore, recommended that the authors perform some additional checks, including clinical isolations and extended genomic/metagenomic analysis, to verify the detection of A. baumannii. Additionally, since there isn't much literature providing evidence of such superbugs present in healthy individuals in vaginal or any other internal microbiome, added to the fact that these pathogens are often nosocomial, the authors should ensure that this detection is not due to any contamination or nosocomial infection at the hospital/s where the samples were collected.
- doi.org/10.3389/fmicb.2021.714284
- doi.org/10.1016/j.synbio.2021.04.002

2. Some studies have targeted an analysis like what the authors have performed here (which essentially refutes some of the claims made by the authors in lines 81—83.) Some of these works find traces of Acinetobacter spp. and have remarked that vaginally born infants may have Acinetobacter spp. infections. Again, such works are far and few. It is imperative that the authors contrast their findings against such published works:
- doi.org/10.1016/j.rbmo.2023.02.002
- doi.org/10.3390/pathogens10020090
- doi.org/10.1007/s10815-021-02247-5
The authors need to better compare their findings against other recent works analyzing the vaginal microbiome. Studies have reported changes to be potentially attributed to conditions like PCOS and other abnormalities in the vagina, alongside changing pH during menstruation which influences the changes in vaginal microbiota throughout the stages of the menstruation cycle.

Additional comments

The authors need to work on their Introduction section. In several paragraphs, the authors have provided a sudden dump of information with little to no context leading up to why this information is provided there, and nothing much succeeds such information in the corresponding paragraphs. For instance, the Lactobacillus CSTs on lines 59—61.

In the Results section, the authors have made some qualitative remarks, such as 'was relatively lower' (line 162), and 'was obviously higher' (line 163). The authors should provide specifics about the observed values or quantities and refrain from making such qualitative remarks in the results.

Reviewer 2 ·

Basic reporting

The material has taken a lot of effort to prepare, but I still feel that the language used in this manuscript is very ambiguous. Literature references and other background is given enough in the correct context.
The architecture of the manuscript is not very strong. Authors have to rewrite the whole abstract again that make clear and concise statements in their abstract that readers can follow. I personally did not find any new findings in this paper.
They have only referred to the published literature as a reference to make sure that their data aligns with the published literature.
It is also not clear why the studies were conducted. It is also unclear in the manuscript what type of experiments they performed except for the Vaginal swabs collection.
The sample size is tiny so I must recommend authors have a bigger sample size and then submit their manuscript in order to be considered with some new findings.

Experimental design

Unfortunately, the aims and research do not have a defined research question that they are addressing. I cannot find the experimental section relevant to the title of the paper and their findings are not relevant to new scientific findings.

Validity of the findings

I cannot find any novelty in this manuscript.

Additional comments

My suggestion to the authors :
1. collect more sample sizes to solidify their findings. I suggest that authors write a manuscript with better structure and that it should be easy to follow.
2. Authors are required to explain their motivation to conduct this study and how it will benefit the scientific community.
3. Authors have worked hard to write the manuscript but I want authors to bring more additional novel findings to add value to the scientific community with their manuscript.
4. I would like to see how these microbiome studies will help in providing information about women's health. Some information that benefits the health of women or is associated with women's health and how the bacterial commensalism will be beneficial?
5. Authors can also focus on the mechanism/ pathway of the presented microbiome and their mode of action in stopping the infection of other bacteria.
6. they can conduct a comparison study between women who are taking antibiotics vs not taking antibiotics. or have another change in the vaginal microbiome.
7. Authors can focus so many other aspects to compare and provide a real metagenomics study after studying the huge number of participants.
I hope this suggestion will help authors to think critical before conducting future experiments and think that how they can provide some novel information in their manuscript.

Reviewer 3 ·

Basic reporting

Overall, the authors have attempted to provide insights into the vaginal microbiome composition in healthy females. The manuscript is well thought out with clear conceptualization and written well. The manuscript has sufficient background information and the aims of the study are much needed to understand the microbiome characterization. The sample collection and size need additional information. The results of the study are very generalizable and need to be validated across different study settings.
Study Limitations needs to be added.

Experimental design

Sample size calculation is required and how did they choose only 14 volunteers?
Participant age: what was the mean age of the volunteers 20-30 years is a wide age range. Patient and clinical characteristics will be essential to understand the differences of the study population.
Participant recruitment: if the study population was recruited from Sichuan University and West China Hospital, how they were confirmed healthy? Did they have any other comorbidities?
Sample collection: Whether 3 samples for the HPVP and 3 samples for MV as per their respective follicular phase, periovulatory phase, and Luteal phases?
As the menstrual cycle differs and the phases are not uniform among women, did the volunteers have clinical screenings to confirm the phases?
Since the samples were collected by participants, how to confirm the swabs for HPVP and MV were accurately sampled from the intended location?
How many samples were collected from Chengdu, Hangzhou, and Shenzhen to study the regional differences in the vaginal microbiome? Did the samples were collected in the same fashion as the current study? The publicly available datasets from Shenzhen and Hangzhou, China were also healthy women.
Overall the experimental design has some weaknesses and needs further clarification and revision.

Validity of the findings

Since the experimental designs require clarification, the results of the finding especially the comparison of public datasets from other provinces need clarification. Not sure, the authors are reporting valid comparisons.

Additionally, the samples were collected by participants and the location of the sample is very essential for the study design and findings. The timing of collection as per the Menstrual cycle phase is also important to confirm the reported findings.

As some of the samples were drop outed for various reported reasons, the sample size and availability is further decreased.

The authors mention diet and lifestyle, but there are no details provided for the impact of these factors in the manuscript. The conclusion says about the different vaginal preparations and requirements for customization as per the province is an application and requires further validation.

---

## Round 0.2 · Minor Revisions

The authors have addressed the concerns raised by the reviewers adequately and have improved the quality of the manuscript significantly. However, few minor concerns need to be addressed before being considered for publication.

1. There are still grammatical and typo errors throughout the manuscript.

2. More emphasis is needed on the novelty or significance of the current study.

3. The introduction still needs extensive revision. Authors are suggested to include more information on the existing knowledge on the topic, gaps in the available literature, rationale, and limitations of the current study.

**Language Note:** The Academic Editor has identified that the English language must be improved. PeerJ can provide language editing services - please contact us at copyediting@peerj.com for pricing (be sure to provide your manuscript number and title). Alternatively, you should make your own arrangements to improve the language quality and provide details in your response letter. – PeerJ Staff

·

Basic reporting

I would like to thank the authors for being receptive to the review comments from all the reviewers and attempting to incorporate the feedback in their manuscript. Although some of the issues raised in the previous review have been resolved in the current manuscript, I believe the current manuscript still has quite a few concerns and issues that need to be resolved before the manuscript is considered for publication.
As pointed out by other reviewers as well, the authors need to rethink their manuscript structuring and positioning in terms of the key contributions in the context of the limitations in the current literature and the need to address those gaps.

Experimental design

1. In an attempt to revise the manuscripts to incorporate the review feedback, the authors have removed lines/paragraphs defining the abbreviations. For instance, CST, HPVP, and MV are not formally defined prior to their use. If an abbreviation is not a de facto standard in the field, it should be defined before its use.
2. It is unclear what are the limitations or gaps in the existing literature that warrant the need for this manuscript/research work. The authors should rewrite the Introduction section expanding the paragraphs on the existing knowledge on vaginal microbiome using amplicon-based and shotgun metagenomic sequencing, follow-up it up with the limitations in the literature and the need to fill this gap, and then provide an overview of their proposed work.
3. The manuscript refers to just 50 publications and 23 of these are from 2015 or before. For a research work using shotgun metagenomics for vaginal study, one of the increasingly studied topics using a microbiome analysis technique used in several studies, the manuscript lacks both quantity and quality (recent relevant works) of reference works cited.

Validity of the findings

As pointed out by other reviewers as well, the novelty or the key contributions of this work are unclear. The authors need to highlight the limitations or shortcomings in the existing literature better and provide context about how their work fills this gap in understanding.

Some of the noticeable findings pertaining to pathogenic bacteria detection correspond to lowly abundant microbes.
The Introduction lines 109--111 read "Due to low resolution and limited functional analysis at the species level, studies based on 16S rRNA amplicon sequencing technology have some limitations". Although the comment about limited functional analysis is true for 16S rRNA amplicon sequencing, the low-resolution issue affects shotgun metagenomic sequencing much more.
Without a detailed genomic analysis that provides a convincing basis (breadth and depth of reference genome coverage, % sequence identity, etc.), the detection of bacteria with low abundance and low genome coverage needs to be dealt with caution while making any major inference.

Additional comments

There are some grammatical errors and typographic errors in the manuscript.

Reviewer 2 ·

Basic reporting

The Authors have edited the manuscript then its previous version. The structure looks professional and the English is easy to understand and to be followed.

Experimental design

no comment

Validity of the findings

The findings are valid and their way of collecting the samples meets the ethical requirements. Conclusions are well stated, linked to original research question.

Reviewer 3 ·

Basic reporting

The authors have addressed the issues raised by the reviewers. The manuscript quality has been significantly improved. The manuscript is well-written with increased clarity. I recommend the manuscript to be accepted in the current form and proceed to the next steps in publication

Experimental design

The authors have updated the experimental design with additional clarity.

Validity of the findings

The only drawback is the sample size and number of samples collected for final analysis. I recommend the authors continue this research work and report any differences from a larger sample size and age group to understand how the vaginal microbiome plays an important role in women's health.

Additional comments

NA

---

## Round 0.3 · accepted · Accept

The authors have adequately addressed the reviewer's comment and the manuscript could be accepted for publication.